# Diagnosis and Treatment of Inflammatory Pseudotumor with Lower Cranial Nerve Neuropathy by Endoscopic Endonasal Approach: A Systematic Review

**DOI:** 10.3390/diagnostics12092145

**Published:** 2022-09-03

**Authors:** Sheng-Han Huang, Chi-Cheng Chuang, Chien-Chia Huang, Shih-Ming Jung, Cheng-Chi Lee

**Affiliations:** 1Department of Neurosurgery, Chang Gung Memorial Hospital at Linkou and Chang Gung University, Taoyuan 33382, Taiwan; 2Division of Rhinology, Department of Otolaryngology, Chang Gung Memorial Hospital, Taoyuan 33382, Taiwan; 3School of Medicine, Chang Gung University, Taoyuan 33301, Taiwan; 4Graduate Institute of Clinical Medical Sciences, College of Medicine, Chang Gung University, Taoyuan 33301, Taiwan; 5Department of Pathology, Chang Gung Memorial Hospital at Linkou and Chang Gung University, Taoyuan 33382, Taiwan; 6Department of Biomedical Engineering, National Taiwan University, Taipei 10617, Taiwan

**Keywords:** lower cranial nerve neuropathy, inflammatory pseudotumor, skull base

## Abstract

Inflammatory pseudotumor (IPT) is a non-neoplastic condition of unknown etiology. IPT with lower cranial nerve (CN IX, X, XI, XII) neuropathies is extremely rare. In this study, we systematically reviewed all previously reported cases regarding the management of IPT with lower cranial nerve neuropathies. We searched the PubMed/MEDLINE database for reports related to IPT with lower cranial nerve neuropathies. A total of six papers with 10 cases met the inclusion criteria (mean age 51.6 years, 70% male). The mean follow-up period was 15.4 months (range: 1–60 months). The most frequent treatment was corticosteroids alone (60%), followed by surgery alone (20%), and multimodal treatment (20%). Corticosteroid therapy was associated with an excellent (100%) response rate at 6 months of follow-up, and half of the patients were in complete remission after 9 months. Both cases who received surgery had persistent neurological deficits. Immediate complete remission of neurological symptoms and resolution on imaging after decompression surgery via an endoscopic endonasal approach (EEA) with corticosteroids was demonstrated in our representative case. This review suggests that EEA is a preferred method for diagnosis and treatment, a promising approach associated with favorable outcomes, and a viable first-line treatment for selected cases, followed by multimodal therapy.

## 1. Introduction

Inflammatory pseudotumor (IPT), also known as plasma cell granuloma or myofibroblastic pseudotumor, is a rare non-neoplastic chronic inflammatory lesion of unregulated inflammatory cell growth, and it is often misdiagnosed as an infection or neoplasm [1,2]. The underlying etiology and pathophysiology of IPT remain unclear [3]. It has been reported at various sites, including the lungs, lymph nodes, orbital cavities, head, neck, and soft tissues; however, it has rarely been reported in the skull base [4]. Furthermore, IPT with lower cranial nerve (CN IX, X, XI, XII) neuropathies is extremely rare. It is a benign lesion associated with various radiographic findings, is clinically site-dependent, and tends to be progressive. 

Complete surgical excision is considered to be curative for lesions located in accessible regions, such as the lungs or gastrointestinal tract [5,6,7]. Steroids or conservative treatment are favored for IPT in inaccessible regions, lesions with high morbidity, or those located at an anatomic location precluding complete resection, such as the orbit, cavernous sinus, or brain [8,9,10]. Surgical resection, steroids, radiation, or a combination of these treatments have been used for IPT of the skull base [11,12]. However, for steroid-resistant or surgically inaccessible skull base lesions, treatment is challenging and controversial, especially for patients with lower cranial nerve (LCN) palsy. This study is the first and most comprehensive systematic review of reported cases of skull base IPT. We focused on cases with LCN neuropathy, and aimed to identify trends in treatment modalities and patient outcomes in the very limited number of reported cases.

## 2. Materials and Methods

This review conforms to the “Preferred Reporting Items for Systematic Reviews and Meta-Analyses” (PRISMA) statement [13]. We searched the electronic database of PubMed/MEDLINE for reports published between January 1990 and December 2021. A language restriction was applied to include articles only in English. The search strategy used the following keywords: [inflammatory pseudotumor] AND [lower cranial nerve neuropathy OR skull base OR clivus]. Abstracts were reviewed independently by two authors, and relevant articles were evaluated. The exclusion criteria were review articles, studies not in English, studies on IgG4-related diseases, Rosai-Dorfman, pseudotumor cerebri, meningioma, and those only reporting extracranial tumors. Each report of IPT with LCN neuropathies was recorded, and data including patient demographics, presented symptoms, tumor location, treatment modality, outcomes, and complications were extracted.

## 3. Illustrative Case

This patient was a 48-year-old man with a history (>20 years) of tinnitus and bilateral hearing impairment. On presentation, he complained of a progressive headache localized to the occipital area and the posterior region of the neck, swallowing difficulties, hoarseness, and limited tongue movement. A physical examination revealed an intolerable headache (numerical rating scale: 8 out of 10), dysphagia, dysarthria, left-sided shoulder pain and weakness (grade 4), and deviation of the tongue toward the left side. Laboratory tests revealed leukocytosis (12,400/μL), and mild elevations in C-reactive protein level (9.8 mg/L) and erythrocyte sedimentation rate (28 mm/h). Serum immunoglobulin G4 (IgG4) was negative. A cerebrospinal fluid (CSF) examination was performed, which did not show any evidence of central nervous system infection; however, the protein content was high (total protein 118.2 mg/dL), suggestive of an inflammatory process. A computed tomography (CT) scan revealed a hyperdense, well-defined lesion with homogeneous enhancement, located posterior to the clivus and ventral to the medulla. Magnetic resonance imaging (MRI) further demonstrated that the mass had infiltrated the retropharyngeal space. Encasement of the clivus bone, internal carotid artery (ICA), jugular foramen (Figure 1, asterisk), and hypoglossal canal was also observed. The unusual presentation (intolerable sharp pain and rapidly progressing neuropathies), and the location of the mass outside the dura mater (Figure 1, arrow), led us to suspect that the underlying cause was a malignancy or an inflammatory process rather than meningioma. A tissue biopsy and decompression of the neuroforamen were performed using an endoscopic endonasal approach (EEA) under navigation guidance (Figure 2, black dashed line). Two surgical specimens were obtained intra-operatively, and neuropathology revealed an absence of neoplastic cells. The final pathology showed fibrosis and chronic inflammation (Figure 3). A CT scan 1 week after the operation showed a well-decompressed hypoglossal canal (Figure 4, asterisk). The patient reported an immediate and dramatic improvement in the headache and cranial nerve neuropathies following surgery, and the subsequent administration of low-dose steroids for 100 days. Serial follow-up MRI only demonstrated post-operative changes (Figure 5, arrows) and did not reveal any evidence of recurrence. In addition, the patient did not report any recurrence of his symptoms.

## 4. Results

### 4.1. Demographics and Clinical Presentation

The systematic search resulted in 73 potentially relevant articles (Figure 6). After screening the abstracts, 67 full-text papers were retrieved and examined for eligibility. A total of six papers was selected for further data extraction. These six studies included 10 patients (7 males and 3 females) with pathologically proven IPT who presented with LCN neuropathies. The demographic and clinical characteristics of these cases are summarized in Table 1. The mean age at diagnosis was 51.6 years and 70% of the patients were male. The mean follow-up period was 15.4 months (range: 1 month to 60 months). The most common LCN symptoms and signs were dysphagia (6/10 (60%)), followed by hoarseness (5/10 (50%)), and tongue atrophy (4/10 (40%)). The most commonly involved LCN was CN XII (9/10 (90%)), followed by CN IX (7/10 (70%)) and CN X (7/10 (70%)). CN XI was least affected, with only one case noted in our review.

### 4.2. Diagnostic Tools

As shown in Table 1, all of the patients underwent MRI examinations except for case 3, who had a pacemaker implanted to treat cardiac arrythmia. MRI revealed a T1-weighted hypointense or T2-weighted hypointense lesion in most patients (5/9, 56%). Only one case had a complete specific serologic study, which was positive for EBEA-Ab, EBNA-Ab, and EB-VCAIgG.

### 4.3. Treatment and Follow-Up Outcomes

Endoscope biopsy was performed in three cases, punch biopsy in two cases, CT-guided biopsy in one case, and transmastoid biopsy in one case. The most common treatments were corticosteroids alone (5/10 (50%)), surgery alone (2/10 (20%)), and multimodal treatment (2/10 (20%)) (Table 1). For those treated with corticosteroids alone, 50% (3/6) had complete remission of neurological symptoms, and another 50% (3/6) had partial improvement (Table 2). In addition, the response rate to corticosteroid therapy was 100% (6/6) at 6 months of follow-up, and half of the patients had complete remission at 9 months of follow-up. Both cases who received surgery had persistent neurological deficits at 2 years and 5 years of follow-up, respectively. The patients who received corticosteroid plus radiation therapy or surgery plus radiation therapy also had persistent disease. Radiographically, follow-up MRI after open surgery showed that one of the two patients had complete resolution, and the other had partial remission or stable disease. In our representative case, complete resolution of the neurological symptoms and lesion in follow-up imaging after EEA decompression surgery and oral corticosteroid therapy were achieved after 3 years of follow-up.

## 5. Discussion

IPT is a pathologic term describing a rare, non-neoplastic chronic inflammatory process. It has been reported to occur at many sites of the body, but rarely in the skull base [2,3,14]. Furthermore, IPT with LCN neuropathies is extremely rare, and consequently the previously published literature is limited to case reports and small case series. To the best of our knowledge, this study is the first systematic review of IPT with LCN.

### 5.1. Etiology

The pathogenesis of IPT remains unclear [19], although it may be due to infectious diseases, autoimmune inflammatory diseases, or the over-production of fibrogenic cytokines [18]. Chang et al. reported that IPT may be associated with a number of disease processes, including Sjögren’s disease [20], Epstein–Barr virus [21,22], human immunodeficiency virus [23] and neuro-Behçet’s disease [24]. Al-Sarraj et al. [19] reported that IPT is the result of an exaggerated immunological process, as supported by increased serum immunoglobulin levels [25]. Viral infection is thought to be a cause of the process because Epstein–Barr virus has been associated with up to 40% of IPT cases [21]. In other cases, the immunological response may be due to an autoimmune disease. 

### 5.2. Demographics and Clinical Presentation

The mean age at diagnosis in the present review was 51.6 years, which is older than that reported in patients with IPT involving the lateral skull base (39.4 years) [3] and sinonasal and ventral skull base (46.7 years) [26]. In our review, 70% of the patients were male, which is higher than in patients with lateral skull base IPT (56.4%) [3] and sinonasal and ventral IPTs (60.9%) [26]. The most common LCN symptom was dysphagia (60%), compared with hearing loss in 53.8% of patients with lateral skull base IPT [3] and vision change in 58.6% of patients with sinonasal and ventral skull base IPT [26], followed by hoarseness (50%), and tongue atrophy (40%). The most common LCN involved was CN XII (90%), compared with CN VII (31.6%) in patients with lateral skull base IPT [3] and CN VI (44.4%) in patients with sinonasal and ventral IPT [26], followed by CN IX (70%) and CN X (70%). CN XI was the least affected, with only 1 case noted in our review. 

Our case was a 48 year-old male patient who reported long-term headache, tinnitus and hearing impairment. He also presented with the most common LCN signs of dysphagia, hoarseness and tongue atrophy. Because of the anatomical proximity to the E-tube and skull base, a longstanding infection in adjacent areas, such as chronic otitis media or chronic sinusitis, may have contributed to the development of IPT of the skull base. The long-term tinnitus and hearing impairment may also be suggestive of a chronic infection.

### 5.3. Diagnostic Tools

Although the diagnosis of IPT is made by exclusion, typical radiographic findings as well as a tissue biopsy remain the gold standard for a definite diagnosis. MRI and CT are the the preferred imaging modalities for IPT with skull base involvement. Although CT can delineate surrounding bone structures with sufficient quality, especially for skull base IPTs, MRI remains the best option, as it can discriminate different kinds of tissue signals. Typically, MRI shows an IPT as a hypointense to isointense abnormality on T1-weighted images, and as a hypointense abnormality on T2-weighted images with homogeneous enhancement [21,22,23]. Variable contrast enhancement has been reported [27], and the MRI findings may resemble the typical features of nasopharyngeal carcinoma, lymphoma, meningioma, chordomas, and other inflammatory or infectious diseases such as sarcoidosis, vasculitic processes, Wegener granuloma and Langerhans histiocytosis in children [18]. IPT can also be difficult to differentiate from meningiomas, because both lesions are generally homogeneously enhancing, and are often associated with the meninges. After cautious interpretation of the MRI findings, an extradural location of IPT can be identified pre-operatively, as in our case. In addition, hypointensity on T2-weighted MRI can help to differentiate IPT from other lesions of the skull base that appear iso- or hyperintense on the same imaging sequence, including chordomas, chondrosarcomas, nasopharyngeal carcinomas, and metastatic malignant tumors [21]. Nevertheless, MRI alone is insufficient to make a definitive diagnosis of IPT. Since IPT involving the skull base arises mostly from dural and meningeal structures [19], tends to be aggressive, and tends to mimic malignant neoplasms or infection, a biopsy to rule out malignancy is imperative. Open or endoscopic biopsy is the preferred method to obtain tissue, and it was performed in 7/9 cases (77.8%) in the present review, compared with 90.8% in patients with sinonasal and ventral skull base IPT [26]. In our case, due to the anatomical location, a complete resection was challenging; therefore, we used an EEA for the partial decompression of the LCNs and obtained a tissue biopsy.

### 5.4. Treatment

With respect to the treatment of skull base IPT, there is currently no consensus due to the limited number of published cases. However, the treatment strategy should be based on the involved site and structures. Accordingly, the ideal treatment algorithm should maximize the extent of tumor resection and neurologic decompression, while minimizing the risk of complications. Corticosteroids, surgical resection, radiotherapy, or a combination of these modalities are the mainstays of therapy for IPT of the head and neck [28]. Corticosteroid therapy should be given if complete surgical resection of the lesion is impossible. Although corticosteroids are fast-acting with an excellent response rate of approximately 80%, the rate of complete remission is only 40–50% [29]. In addition, recurrence after the cessation of corticosteroids occurs in approximately 20% of patients with IPT of the parapharyngeal space or skull base [30]. In our review, corticosteroid therapy resulted in a 100% response rate (complete resolution and partial improvement) to IPT with LCN involvement without recurrence. It takes an average of 6 months for patients to respond to corticosteroids alone. Since the follow-up period was relatively short, a study with longer follow-up is important for clinical practice.

Considering the anatomical complexity and proximity to critical neurovascular structures, the resection of skull base tumors involving the lower clivus remains a challenge. Approaches including traditional craniotomy, such as retrosigmoid, far lateral routes or modified skull base approaches may provide an adequate surgical corridor to this area; however, this necessitates excessive, time-consuming bony work and long cerebellar retraction times. The risk of post-operative morbidity due to cerebellar injury and LCN palsies (CN IX-XII) is high, because these nerves are situated on the surface of the lesion if approached from the posterior via craniotomy. The results of our systematic review show that the use of a craniotomy approach for tumor resection increases the risk of post-operative complications, and may provide limited outcome benefits.

With a further understanding of the skull base anatomy, advances in surgical techniques, and the use of navigation guidance, EEA has become the most popular approach in the management of skull base lesions. EEA not only provides wide exposure and clear visualization of the lesion from below, but also minimizes bone destruction, brain retraction, and nerve injury. EEA is not as minimally invasive as the biopsy procedure; however, with great advances in this approach and collaboration with otorhinolaryngologists, EEA may not only minimize destruction, but also promise rapid recovery. In our case, MRI clearly demonstrated an extradural lesion, and therefore EEA was the best approach to obtain tissue samples and achieve decompression without the risk of nerve damage or CSF rhinorrhea. The most common complications of EEA are sinusitis, anosmia and empty nose syndrome. With the assistance of otorhinolaryngologists, the olfactory epithelium and mucosa were well-preserved, and the destruction was minimized, which diminished the above complications. Based on our review (Table 1 and Table 2), we propose that minimally invasive EEA can be considered the first-line treatment for selected cases, and that corticosteroids can be considered safe and effective adjuvant therapy for skull base IPT. As mentioned above, although the location in our case was surgically accessible via EEA, it was impossible to perform gross total resection due to bone and ICA involvement and consequent concerns over safety. In cases where only a punch biopsy is performed, the post-operative improvement would take longer, and the patient would need to take a higher dosage of corticosteroids for a longer period of time. Furthermore, to improve the recovery of the involved LCNs, partial resection with subsequent corticosteroid treatment is the best policy, compared with biopsy only or radical resection. Our case showed an immediate and dramatic improvement in cranial nerve neuropathies after EEA decompression surgery and short-term corticosteroid treatment. Despite significant bony resection around the jugular foramen and hypoglossal canal, the patient did not report any relevant symptoms, such as headache, neck pain or muscle rigidity.

For patients who do not undergo a biopsy and take corticosteroids alone, high-dose steroid therapy followed by low-dose steroid maintenance therapy (LDSMT) is widely used as the initial treatment to prevent relapse. However, the complete resolution of IPT has not been shown on imaging studies after this approach [17]. Long-term LDSMT may be a treatment option in steroid-dependent patients. However, well-known complications of steroid therapy include cutaneous atrophy, osteoporosis, and the transient aggravation of glucose intolerance in patients with diabetes. Therefore, the efficacy and safety of long-term LDSMT in patients with IPT still remains questionable. Partial decompression via EEA may not only expedite the recovery of neurological deficits, but also reduce the period of steroid usage and related long-term complications. In our case, the LDSMT dosage after EEA was prednisolone 10 mg/day, lasting for about 100 days. With this approach, the patient showed a dramatic and persistent improvement, and complete remission of his neurological deficits without long-term complications. Therefore, a minimally invasive approach for partial resection followed by LDSMT appears to be a suitable treatment strategy.

Radiation therapy is a reasonable alternative option for cases where surgery is not possible or patients are unresponsive to steroids. However, low-dose radiation therapy has not been reported to achieve complete remission [31]. Doses between 2000 and 4000 rad have been associated with a positive clinical response for intracranial IPT [32]. Chemotherapy has not been found to be effective to date [33]. Therefore, we propose that multimodal treatment with EEA decompression surgery followed by corticosteroid and/or high-dose radiation therapy can be considered the first-line treatment for selected cases as the most reasonable approach with the most effective favorable outcomes. We recommended adopting this approach as the first-line treatment in the following situations: (1) patients did not respond well to the initial steroid treatment no matter whether biopsy was done or not; (2) complications occurred after long-term steroid treatment; (3) patients were immunocompromised or had underlying co-morbidities, such as DM, osteoporosis or infectious process, which precluded the long-term use of steroid. Based on the literature review and our clinical experience, we propose a diagnostic and therapeutic algorithm for suspicious skull base IPT with lower cranial nerve neuropathy (Figure 7).

### 5.5. Follow-Up Outcomes

The mean follow-up period was 15.4 months in the present review, compared to 21.6 months reported in patients with lateral skull base IPT [3] and 17.6 months in those with sinonasal and ventral skull base IPT [26]. Due to the surgical challenges with IPT located near the skull base foramen and LCNs, it is reasonable that the most common treatments focus on corticosteroids alone (60%). Of the patients treated with corticosteroids alone, 50% showed complete remission of neurological symptoms, and another 50% showed partial improvements. In addition, the response rate to corticosteroid therapy was 100% (6/6) at 6 months of follow-up, and half of the patients showed complete remission at 9 months of follow-up. However, a recurrence of symptoms was found in most patients after stopping steroid therapy. Imaging studies also showed the reactivation of disease in many patients. Twenty percent of the patients received surgery alone without subsequent steroid or radiation therapy. Another 20% received multimodal treatment (surgery or corticosteroids followed by radiation therapy) without any significant improvement thereafter. Both cases who received surgery had persistent neurological deficits at 2 years and 5 years of follow-up, respectively. In the patients who underwent surgery alone, follow-up MRI showed complete or partial remission of the disease. Based on our review, it seems that surgical resection for IPT located at the skull base with LCN involvement carries a high risk of post-operative complications and morbidity. There appeared to be almost no neurological improvement in the patients who underwent surgery (cases 3 and 4). There was an improvement after en block resection in case 7; however, the insidious development of diplopia was found, and oculomotor nerve palsy was confirmed 6 months post-operatively. Furthermore, follow-up MRI revealed local recurrence, and whole brain radiation therapy was conducted. Fortunately, the diplopia regressed gradually and the recurrent tumor diminished in size upon serial MRI scans. No recurrence of the tumor was documented radiographically or clinically after 2 years of follow-up. In the current review, no patients underwent partial resection via a minimally invasive approach followed by corticosteroid treatment, as in our case, wherein the patient had an uneventful post-operative course with complete radiographic and clinical remission. An experienced skull base surgeon who is familiar with the EEA approach can significantly reduce the associated risks. As in our case, complete remission of neurological symptoms and complete resolution of the tumor in follow-up imaging was achieved through EEA decompression surgery with oral corticosteroid treatment at 3 years of follow-up. As a result, for skull base IPT with LCN neuropathy, overly aggressive surgical resection will lead to post-operative neurological deficits and unwanted complications, and hinder patient recovery. In contrast, if a patient undergoes conservative treatment with steroid or radiation therapy only, the physician will be unable to obtain a tissue biopsy, thereby preventing a confirmative diagnosis and precise treatment.

The current study has several limitations. First, all of the included studies were case reports or small case series characterized by a high risk of publication bias toward treatment outcomes. Therefore, the results may not accurately reflect the treatment effects of an average patient with skull base IPT. Second, the length of follow-up was variable, ranging from 1 month to 5 years. Shorter follow-up may risk underreporting complications or recurrence. Lastly, the rarity of the disease prevented us from performing a meta-analysis. Further high-quality evidence is necessary to help guide definitive treatment plans.

## 6. Conclusions

Skull base IPT with lower cranial nerve neuropathy is an extremely rare condition, and an accurate diagnosis is challenging based on clinical presentation and radiographic findings. From our experience and systematic review, a minimally invasive biopsy and decompression with EEA followed by corticosteroid and radiation therapy are the optimal treatment methods.

## Figures and Tables

**Figure 1 diagnostics-12-02145-f001:**
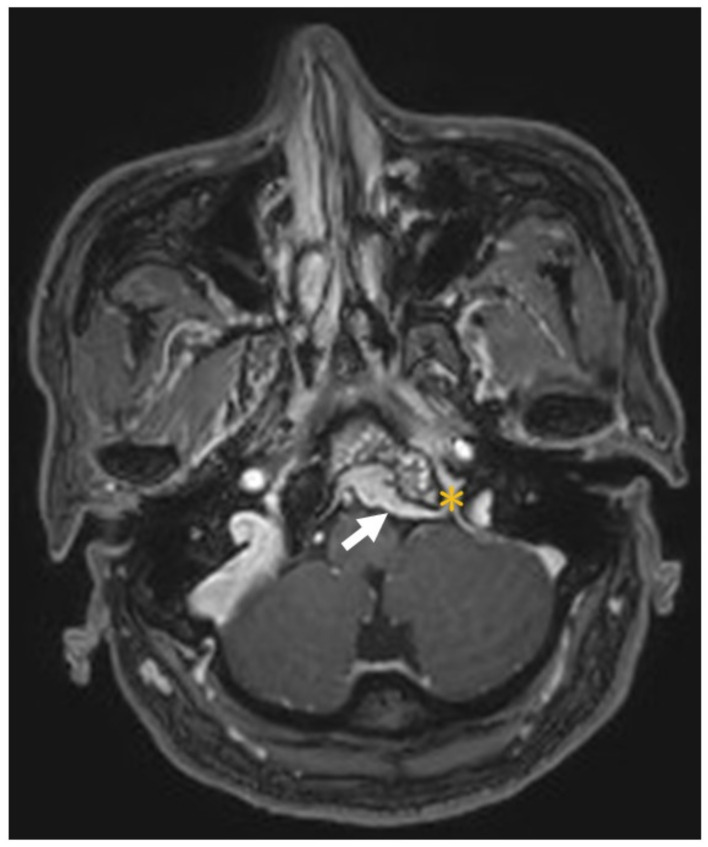
Magnetic resonance imaging of an enhanced lesion in the retropharyngeal space with encasement of the clivus bone, internal carotid artery, jugular foramen (asterisk) and hypoglossal canal (not shown). The lesion was located outside the dura mater (arrow).

**Figure 2 diagnostics-12-02145-f002:**
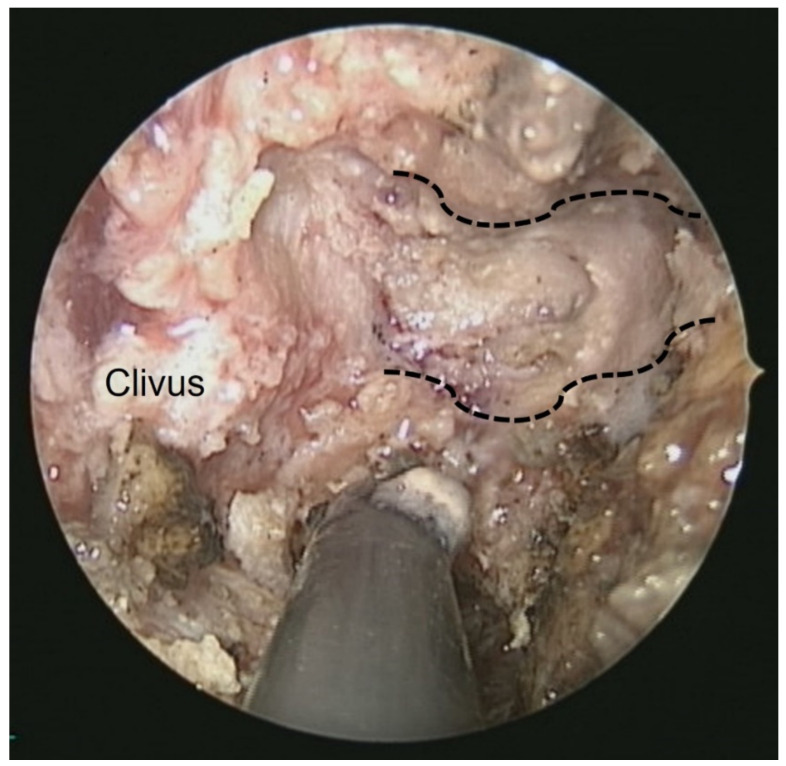
Intra-operative endoscopic view from the nose: the left side hypoglossal canal (black dashed line) and underlying lower cranial nerves were well-decompressed.

**Figure 3 diagnostics-12-02145-f003:**
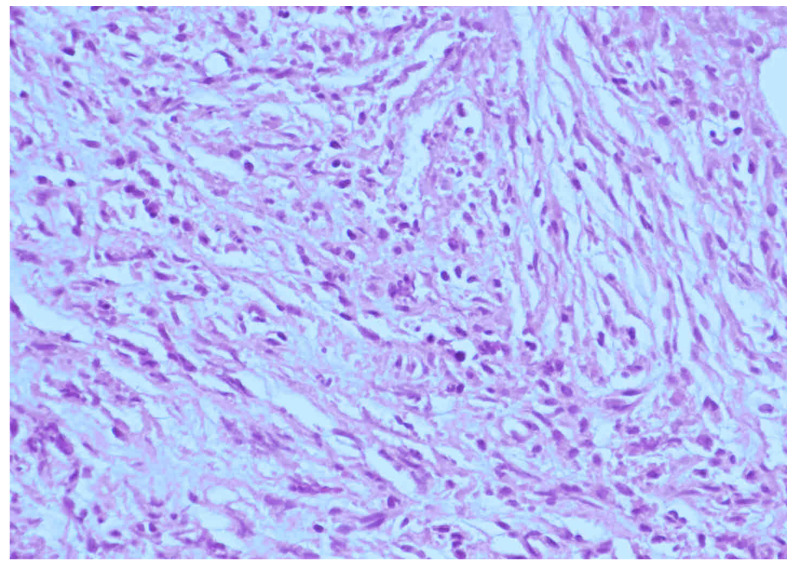
Histological section of the surgical specimen showing acute and chronic inflammatory changes, including inflammatory cells with fibrosis (hematoxylin and eosin staining, original magnification 400×). A diagnosis of an inflammatory pseudotumor (IPT) was made.

**Figure 4 diagnostics-12-02145-f004:**
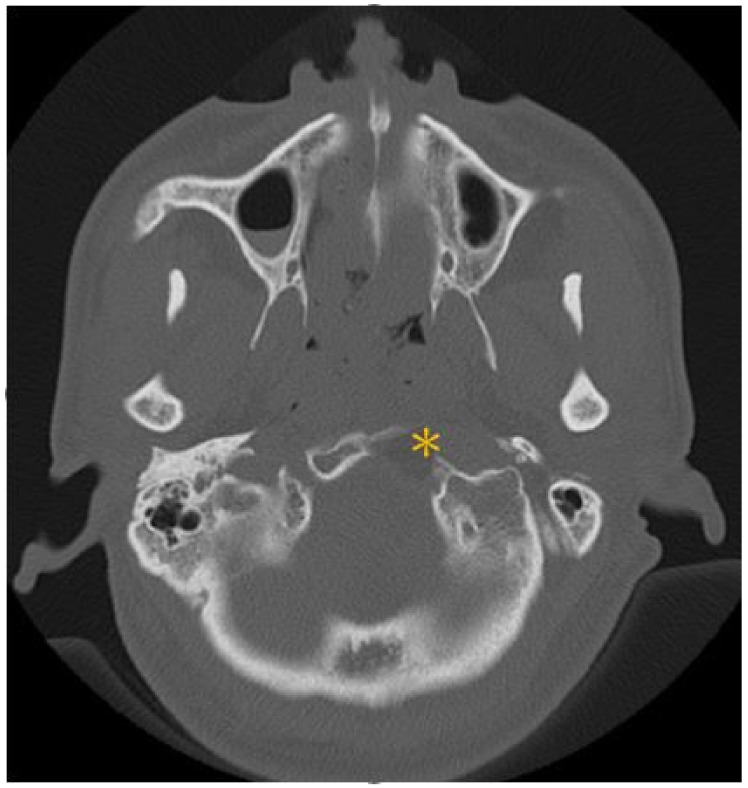
Post-operative CT scan disclosed a well-decompressed hypoglossal canal (asterisk).

**Figure 5 diagnostics-12-02145-f005:**
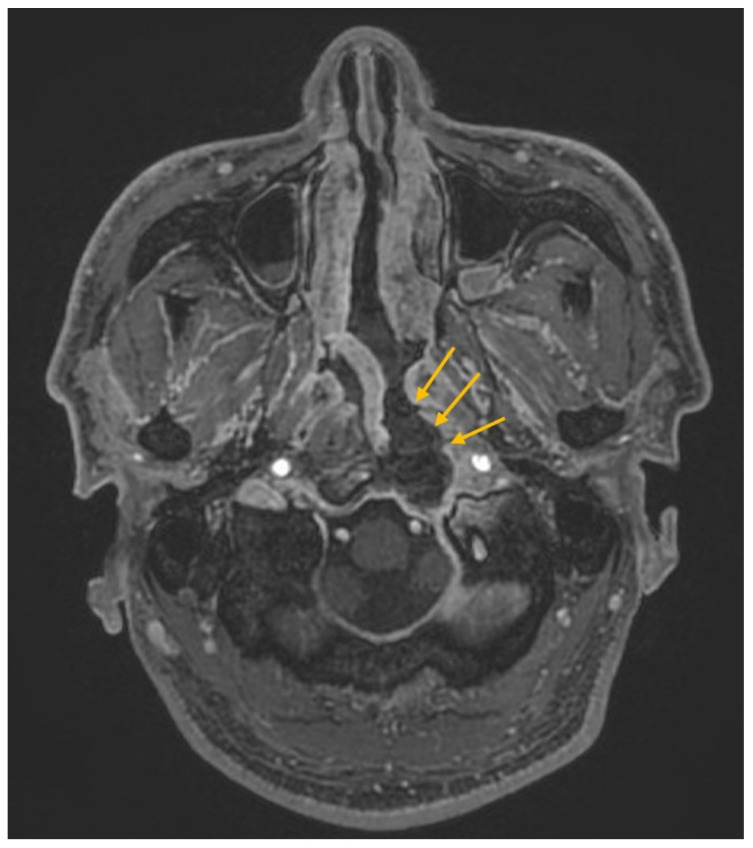
Post-operative follow-up MRI only demonstrated post-operative changes (arrows) and did not reveal any evidence of IPT recurrence.

**Figure 6 diagnostics-12-02145-f006:**
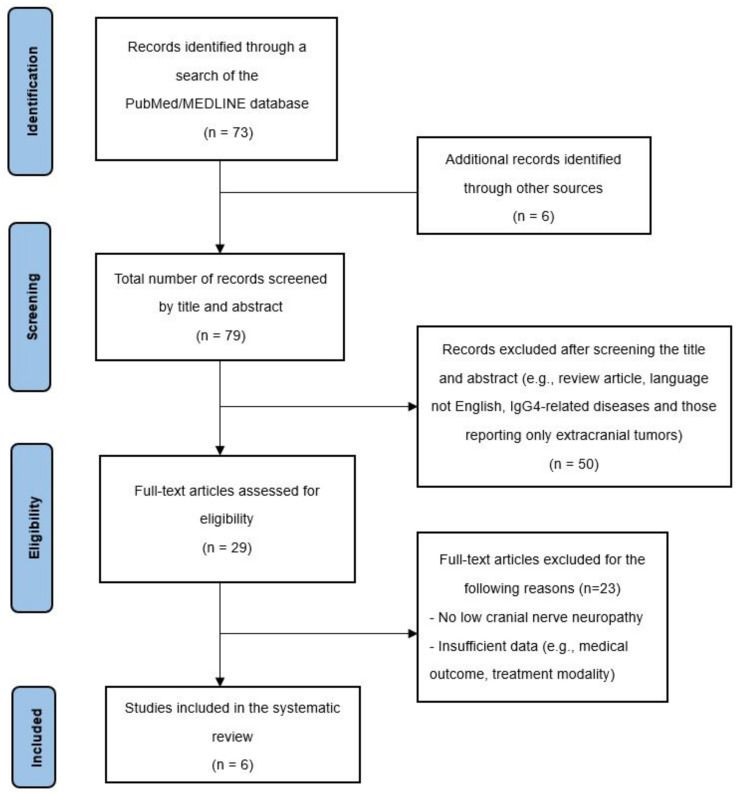
PRISMA diagram describing the case selection process.

**Figure 7 diagnostics-12-02145-f007:**
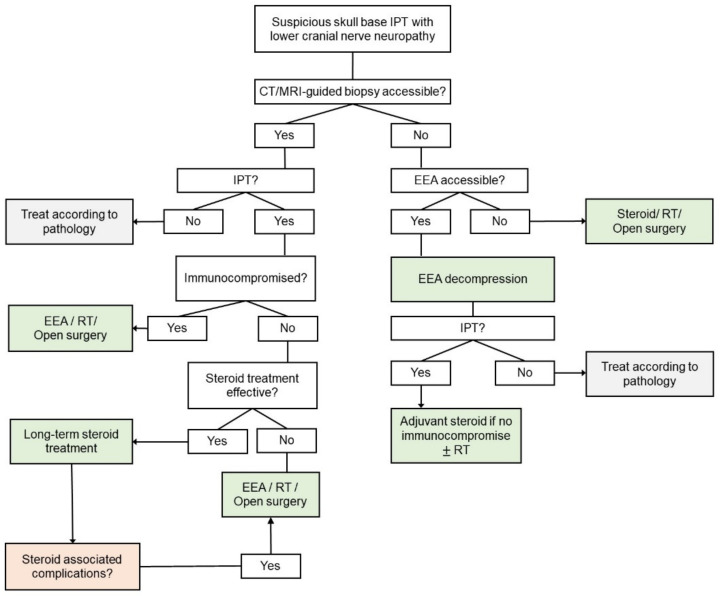
Proposed diagnostic and therapeutic algorithm for suspicious skull base IPT with lower cranial nerve neuropathy. Abbreviations: CT = computed tomography; EEA = endoscopic endonasal approach; IPT = inflammatory pseudotumor; MRI = magnetic resonance imaging; RT = radiation therapy.

**Table 1 diagnostics-12-02145-t001:** Literature summary of demographic characteristics and treatment modality of inflammatory pseudotumor with lower cranial nerve neuropathy.

Year	Author	No.	Age /Sex	Lower Cranial Nerve S/S	Involved Regions	Involved Nerves	MRI Findings	Serology ^1^	Biopsy	Treatment Modality	Follow-up Period
1997	Sung et al. [14]	1	57F	Dysphagia	JF, ITF	VII, IX, X, XII	ND	-	Endoscope	Prednisolone (oral): 60–100 mg/d	6 mo
		2	41M	Dysphagia, dysarthria, hoarseness	JF	IX, X, XII	T1 hypo T2 hypo	-	Endoscope	Prednisolone (oral): 60–100 mg/d	6 mo
2002	Pallini et al. [15]	3	49F	Tongue atrophy	FM, clivus	VIII, XII	None	-	nil	Complete resection surgery	5 y
		4	46F	Dysphagia, tongue atrophy	FM, clivus, brain stem and upper cervical compression	XI, X, XI, XII	T1 hypo	-	nil	Partial resection surgery	2 y
2004	Crovetto et al. [16]	5	72M	Dysphagia, tongue atrophy	NP, clivus, HC	IX, X, XII	T1 iso	-	Endoscope	Prednisolone (Total 552 mg IM for 24 days + 30 mg/day oral for 45 days)	1 y
2006	Lee et al. [17]	6	63M	Dysphagia, hoarseness		IX, X, XII	ND	-	Transmastoid	Prednisolone (oral): 60 mg/day for 28 days + 10 mg/day for 56 days + RT (2000 cGy)	6 mo
2009	Lin et al. [4]	7	49M	Hoarseness, slow gag reflex, uvula deviation,	CPA	IX, X	enhanced	+	nil	En bloc surgery + whole brain RT (1200 cGy in 6 fractionations)	2 y
2010	Lu et al. [18]	8	70M	Dysphagia, hoarseness	NP, HC, JF	II, III, V, VII, IX, X, XII	T1 hypo T2 hypo	-	NP punch	CS (oral)	1 mo
		9	32M	Tongue atrophy	NP, Clivus, HC, JF	XII	T1 hypo T2 hypo	-	CT-guided	CS (oral)	7 mo
		10	37M	Hoarseness	NP, clivus, HC, JF	XII	T1 hypo T2 hypo	-	NP punch	CS (oral)	8 mo
Present case	Huang et al.		48M	Dysphagia, hoarseness, tongue atrophy	JF, Clivus, HC, CA	IX, X, XII	T1 iso T2 hypo	+	nil	Endoscopic decompression surgery + prednisolone (10 mg/day for 100 days, oral)	3 y

^1^ “-” denotes that the author did not mention clearly the specific type of radiographic or serologic examination. Abbreviations: CA = carotid artery; CPA = cerebellopontine angle; CS = corticosteroids; FM = foramen magnum; HC = hypoglossal canal; ITF = infratemporal fossa; JF = jugular fossa; ND = not documented; NP = nasopharynx; S/S = symptoms and signs.

**Table 2 diagnostics-12-02145-t002:** Symptomatic and radiographic outcomes after different treatment modalities for patients with inflammatory pseudotumor with lower cranial nerve neuropathy.

Outcomes	All Patients	Treatment Modality				
		CS Alone	Surgery Alone	CS + RT	Surgery + RT	EEA+ CS
**Symptomatic**						
Complete resolution	4	3 (50%)	0 (0%)	0 (0%)	0 (0%)	1 (100%)
Partially improved	3	3 (50%)	0 (0%)	0 (0%)	0 (0%)	0 (0%)
Persistent	4	0 (0%)	2 (100%)	1 (100%)	1 (100%)	0 (0%)
**Radiographic**						
Complete resolution	5	3 (50%)	1 (50%)	0 (0%)	0 (0%)	1 (100%)
Remission or stable	5	3 (50%)	1 (50%)	0 (0%)	1 (100%)	0 (0%)
Progression or die	1	0 (0%)	0 (0%)	1 (100%)	0 (0%)	0 (0%)

CS = corticosteroids, EEA = endoscopic endonasal approach, RT = radiation therapy. The percentage represents recovery status after individual treatment method.

## Data Availability

Not applicable.

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
