# Peer review of "Diagnosis and Treatment of Inflammatory Pseudotumor with Lower Cranial Nerve Neuropathy by Endoscopic Endonasal Approach: A Systematic Review"

_diagnostics, 2022, doi:10.3390/diagnostics12092145_

Round 1

Reviewer 1 Report

The title of this "care report and review of the literature" study was interesting, and the paper was well written.

Author Response

Point 1: The title of this "care report and review of the literature" study was interesting, and the paper was well written.

Response 1: Thank you for the positive comments and affirmation. We are looking forward to the acceptance and publication of our paper.

Reviewer 2 Report

The present systematic review and report of an additional case of skull base IPT is quite interesting, well written and the review nicely conducted; the authors finally statement is in favor of IPT multimodal treatment with EEA decompression surgery as first-line treatment for selected accessible lesions to confirm the diagnosis, followed by corticosteroid and/or high-dose radiation therapy being in their opinion the most reasonable approach with the most effective favorable outcomes. 

In my opinion what the authors missed to say is that steroids represent the essential management of IPT (especially in skull base and high complex lesion) in line with pertinent literature thus in the clinical decision making process the goal is to achieve the diagnosis; decompressive surgery is a limited tool to be used in very selected case as it is characterized by a high morbidity rate (lots of complication) and instead a straightforward CT or MRI guided biopsy could be the best and less invasive option; finally EEA could not be defined as a minimally invasive approach as it is a quite aggressive and destructive procedure although a scare is not visible (please discuss, add details about complications, explain better your own policy). 

Finally the study could be greatly improved if the authors would add a clinical and decisional  algorithm to guide practitioners in their routine work to choice and tailor the best treatment on a patient by patient base. 

Author Response

Point 1:

The present systematic review and report of an additional case of skull base IPT is quite interesting, well written and the review nicely conducted; the authors finally statement is in favor of IPT multimodal treatment with EEA decompression surgery as first-line treatment for selected accessible lesions to confirm the diagnosis, followed by corticosteroid and/or high-dose radiation therapy being in their opinion the most reasonable approach with the most effective favorable outcomes.

In my opinion what the authors missed to say is that steroids represent the essential management of IPT (especially in skull base and high complex lesion) in line with pertinent literature thus in the clinical decision making process the goal is to achieve the diagnosis; decompressive surgery is a limited tool to be used in very selected case as it is characterized by a high morbidity rate (lots of complication) and instead a straightforward CT or MRI guided biopsy could be the best and less invasive option; finally EEA could not be defined as a minimally invasive approach as it is a quite aggressive and destructive procedure although a scare is not visible (please discuss, add details about complications, explain better your own policy).

Response 1:

Thanks for your precious time and comments. Besides, we also described our perspective about complications and our policy to minimize the risk. We agreed that EEA can be considered first-line treatment for selected cases, and steroid is the principal treatment in IPT involving the skull base. However, based on our review, only 50% of patient had complete resolution after steroid treatment. In addition, long-term steroid usage would result in related well-known complications. Therefore, the efficacy and safety of long-term steroid therapy in patients with IPT still remains questionable. Partial decompression via EEA may not only expedite recovery of neurological deficits but also reduce the period of steroid usage and related long-term complications. EEA + short-term steroid +/- radiation therapy may also be a more suitable option in the following situations: (1) patients who did not respond well to the initial steroid treatment no matter the biopsy was done or not; (2) complications occurred after long-term steroid treatment; (3) patients had underlying co-morbidities, such as DM, osteoporosis or infectious process, which precluded the long-term use of steroid (We also added this part in our revised manuscript, line 300~304).

(Section of 5.4. Treatment, lines 254~256) EEA is not as minimally invasive as the biopsy procedure; however, with great advances in this approach and the collaboration with otorhinolaryngologist, EEA may not only minimize the destruction but also promise the rapid recovery.

(Section of 5.4. Treatment from lines 259-262) The most common complications of EEA were sinusitis, anosmia and empty nose syn-drome. With the assistance from otorhinolaryngologist, the olfactory epithelium and mucosa were well-preserved, and the destruction was minimized, which diminished the above complications.

Point 2:

Finally the study could be greatly improved if the authors would add a clinical and decisional  algorithm to guide practitioners in their routine work to choice and tailor the best treatment on a patient by patient base.

Response 2:

Thank you for the important suggestion. We have added the Figure 7 which is a diagnostic and therapeutic algorithm for suspicious skull base IPT with lower cranial nerve neuropathy. We hope it can give guidance for practitioners in their routine work.

Reviewer 3 Report

The authors reported a case of Inflammatory pseudotumor associated with the lower cranial nerves, and also performed a systematic review of these diseases.

This article may provide clinical insight in these extremely rare diseases.

I recommend the following several corrections.

1. I believe that “lower cranial nerve” is a more common notation than “low cranial nerve”.

2. Please illustrate the anatomical structure of the endoscopic picture (Figure 2) more clearly for the readers. If possible, an illustration would also be helpful.

Author Response

Point 1:

I believe that “lower cranial nerve” is a more common notation than “low cranial nerve”.

Response 1:

Thank you for the important suggestion. We have changed all the “low cranial nerve” to “lower cranial nerve” in our article. 

Point 2:

Please illustrate the anatomical structure of the endoscopic picture (Figure 2) more clearly for the readers. If possible, an illustration would also be helpful.

Response 2:

Thank you for your considerable advice. We have revised the Figure 2, showing intra-operative endoscopic view from the nose: the left side hypoglossal canal (black dashed line) and underlying lower cranial nerves were well-decompressed. We hope the change can make it more clear for the readers.

Round 2

Reviewer 2 Report

The authors answered reasonably well to raised critics, suggestions and proposed changes.